



# University of Colorado and Black Swift Technologies RPAS-based measurements of the lower atmosphere during LAPSE-RATE

Gijs de Boer[1,2], Cory Dixon[3]*, Steven Borenstein[3], Dale A. Lawrence[3], Jack Elston[4], Daniel Hesselius[5], Maciej Stachura[4], Roger Laurence III[3]#, Sara Swenson[3], Christopher M. Choate[3], Abhiram Doddi[3], Aiden Sesnic[3], Katherine Glasheen[3], Zakariya Laouar[3], Flora Quinby[3], Eric Frew[3] and Brian M. Argrow[3]

[1] Cooperative Institute for Research in Environmental Sciences, University of Colorado Boulder, Boulder, Colorado USA
[2] Physical Sciences Division, National Oceanic and Atmospheric Administration, Boulder, Colorado, USA
[3] Department of Aerospace Engineering, University of Colorado Boulder, Boulder, Colorado USA
[4] Black Swift Technologies, Boulder, Colorado, USA
[5] Office of Integrity, Safety, and Compliance, University of Colorado Boulder, Boulder, Colorado, USA

* Current Affiliation: Geotech Environmental Equipment, Denver, Colorado, USA
# Current Affiliation: Ball Aerospace & Technologies Corporation, Boulder, Colorado, USA

*Correspondence to*: Gijs de Boer (gijs.deboer@colorado.edu)

**Abstract.** Between 14 and 20 July 2018, small remotely-piloted aircraft systems (RPAS) were deployed to the San Luis Valley of Colorado (USA) together with a variety of surface-based remote and in-situ sensors, and radiosonde
systems as part of the Lower Atmospheric Profiling Studies at Elevation – a Remotely-piloted Aircraft Team Experiment (LAPSE-RATE). The observations from LAPSE-RATE were aimed at improving our understanding of boundary layer structure, cloud and aerosol properties and surface-atmosphere exchange, and provide detailed information to support model evaluation and improvement work. The current manuscript describes the observations obtained using four different types of RPAS deployed by the University of Colorado Boulder and Black Swift
Technologies. These included the DataHawk2, the Talon and the TTwistor (U. of Colorado) and the S1 (Black Swift Technologies). Together, these aircraft collected over 30 hours of data throughout the northern half of the San Luis Valley, sampling altitudes between the surface and 914 m AGL. Data from these platforms are publicly available through the Zenodo archive, and are co-located with other LAPSE-RATE data as part of the Zenodo LAPSE-RATE community (https://zenodo.org/communities/lapse-rate/). The primary DOIs for these datasets are
10.5281/zenodo.3891620 (DataHawk2, de Boer et al., 2020a), 10.5281/zenodo.4096451 (Talon, de Boer et al., 2020b), 10.5281/zenodo.4110626 (TTWISTOR, de Boer et al., 2020c), and 10.5281/zenodo.3861831 (S1, Elston and Stachura, 2020).

## 1 Introduction

During the summer of 2018, the University of Colorado, along with the Earth System Research Laboratory (ESRL) of the National Oceanic and Atmospheric Administration (NOAA) and National Center for Atmospheric Research (NCAR) hosted the annual meeting for the International Society for Atmospheric Research using Remotely-piloted Aircraft (ISARRA; de Boer et al., 2019a). Along with this conference the organizers set up a "flight week", where participating groups could conduct remotely-piloted aircraft system (RPAS) flights in a coordinated manner to make scientifically relevant measurements of the lower atmosphere. This campaign, titled "Lower Atmospheric Profiling Studies at Elevation – a Remotely piloted Aircraft Team Experiment" (LAPSE-RATE, de Boer et al., 2020a; 2020b) brought together a variety of teams to Colorado's San Luis Valley for a week of atmospheric science-centric RPAS operation. A summary of the LAPSE-RATE campaign is provided in an overview paper at the beginning of this special issue (de Boer et al., 2020b).

As a part of the LAPSE-RATE effort, a team of participants from the University of Colorado Boulder (CU) and Black Swift Technologies (BST) contributed a week of time and RPAS and ground-based equipment. These teams, originating in CU's Integrated Remote and In-Situ Sensing (IRISS) grand challenge project, the Cooperative Institute for Research in Environmental Science (CIRES), and the Department of Aerospace Engineering's Research and Engineering Center for Unmanned Vehicles (RECUV), consisted of a combination of scientists, engineers and students, and leveraged the technology and expertise developed over a variety of previous successful RPAS-based atmospheric science deployments conducted by the University of Colorado. These include development of equipment and execution of field campaigns to better understand the thermodynamic and dynamic environments around supercell thunderstorms (Koch et al., 2018; Frew et al., 2012; Houston et al., 2012), extensive operation of RPAS at high latitudes (de Boer 2019b; de Boer et al., 2019c; de Boer et al., 2018; de Boer et al., 2016; Cassano et al., 2016; Cassano, 2014; Intrieri et al., 2014; Knuth et al., 2013) and the use of these platforms to understand turbulence across the world (Cione et al., 2019). Similarly, BST has an extensive background of conducting RPAS operations for environmental science as evidenced by a variety of studies and synthesis papers (Elston et al., 2015; Elston et al., 2011a; Elston et al., 2011b). This experience has resulted in a variety of innovative operational modes, including a roaming flight pattern called "follow me", in which the aircraft is trained to follow a flight pattern centered on a moving vehicle. This pattern provided insight into spatial gradients across the valley, as discussed later.

For the LAPSE-RATE campaign, the CU and BST teams were interested in contributing to all of the science goals connected to the campaign, including:

- Convective initiation
- The morning boundary layer transition
- Density currents and drainage flows
- Aerosol properties
- Sensor and platform intercomparison work



CU/BST participation included a significant number of participants (>15), several operations vehicles, over 10 unmanned aircraft, and a mobile atmospheric observatory, command/control center, and computing facility (see de Boer et al., 2020f). The next section provides an overview of the RPAS deployed, including information on the
platforms and the sensors they carried, and the unique capabilities offered by each. Section 3 provides insight into the sampling that was completed, including information on measurement locations, the types of weather conditions sampled, general details on flight strategies, statistics summarizing the flights and data collected, and information on interesting events sampled. Section 4 provides an overview of data processing and quality control, while section 5 offers insight into data availability (and DOIs). Finally, we summarize the paper and dataset and provide information
on contributions of authors, funding sources, and collaborators.

## 2 Overview of Platforms and Sensors

During LAPSE-RATE, CU and BST operated a combination of six different RPAS types. These range from
commercially-available airframes with open-source autopilot systems to custom systems entirely designed, developed and constructed at CU and BST. In this section, we provide an overview of the different aircraft and the sensors carried by each.

One of the CU aircraft with the most amount of flight time during LAPSE-RATE was the DataHawk2 platform (Figure
1a). This system, developed and designed entirely within RECUV, consists of a lightweight (< 1 kg take-off weight), 1.2 m fixed-wing foam pusher-prop aircraft, with custom autopilot system and sensor suite. Capable of approximately 45 minutes of flight with a standard battery configuration and featuring a cruise speed of 14-16 m s$^{-1}$, the DataHawk2 has a theoretical range of approximately 40 km. It has operated in a variety of environments, including the high Arctic (de Boer et al., 2018b; de Boer et al. 2019b), Asia (Kantha, et al., 2017), and South America (Scipión et al., 2016).
The DataHawk2's relatively slow flight speed (14 m/s, burst up to 22 m/s) allows the platform to obtain measurements at high spatial resolution when compared to other aerial vehicles making it ideal for studies of atmospheric turbulence at fine scales (e.g. Kantha et al., 2017; Balsley et al., 2018). Despite this relatively slow speed, the DataHawk2 has been operated in winds up to 12 m/s, making it a robust research platform for the harsh Arctic environment. The aircraft is capable of autonomous operation, although most flights (including all of the LAPSE-RATE flights) are
completed under a remote-pilot scenario where the aircraft is guided and monitored using a computer ground station. The DataHawk2 collected 8.84 hours of data during LAPSE-RATE.

The DataHawk2 system carries a variety of sensors for measuring atmospheric and surface properties. Custom-built instrumentation includes a fine wire sensor array employing two cold- and one hot-wire. These fine wires offer high-
frequency (800 Hz) information on temperature and fine scale turbulence. High bandwidth is enabled by small surface-area-to-volume ratios of very thin (5 μm diameter) wires. In addition to the finewire array, the DataHawk2 carries custom electronics that include integrated-circuit slow response sensors (Sensiron SHT-31) for measurement of temperature through a calibrated semiconductor, and relative humidity using a capacitive sensor. To measure surface



and sky brightness temperatures, DataHawk2s are equipped with up- and downward-looking thermopile sensors (Semitec 10TP583T with custom electronics). To provide additional information on thermodynamic structure, DataHawk2s have also carried the commercially-available iMet1 radiosonde package, providing comparative information on position (GPS), temperature (bead thermistor), pressure (piezoresistive) and relative humidity (capacitive), as well as the Vaisala RSS421 sensor that is similar to the sensor suite used for the popular RS-41 radiosonde. However, neither the iMet1 nor RSS421 were installed during LAPSE-RATE. Finally, to estimate winds using the DataHawk2, information from the onboard GPS, Pitot tube and inertial measurement unit (IMU) are integrated to calculate all three wind vectors. This calculation is completed using equations presented in van den Kroonenberg (2008), under the assumption that the aircraft instantly reacts to perturbations in the local wind vector, meaning that the sideslip angle ($\beta$) is assumed to be zero and that the angle of attack ($\alpha$) is set to be equal to the mean pitch of the aircraft in level flight.

In addition to the DataHawk2, CU collaborated with BST to operate a Black Swift S1 RPAS (Figure 1b) during LAPSE-RATE. This platform was originally designed as an advanced, fully autonomous survey and mapping platform. Equipped with the Black Swift Technologies SwiftCore flight management and autopilot system, remote operation and flight planning is done from a tablet. Take-off is performed through non-assisted hand launches, and the advanced landing algorithm provides for robust and precise autonomous belly landings. The aircraft has a 1.7 m wingspan and a gross take-off weight of 2.5 kg. When operated with a 13,600 mAh LiPo battery, it allows for an endurance of up to 1.5 hours at a cruise speed of 17 m s$^{-1}$, depending on the mission profile and environmental conditions. The aircraft has a high operational ceiling and has been used to perform mapping missions at altitudes up to 14,000 feet in Colorado. It has been employed for surveying work, land management, crop damage assessments, and large area ecological studies. During LAPSE-RATE, the S1 was equipped with the Black Swift Technologies Multihole Probe (BST MHP). This probe includes five pressure ports to measure the system's airspeed, angle of attack and sideslip angle that can be combined with information from an INS and GPS to provide estimates of the horizontal and vertical wind speed and direction. The BST MHP includees an integrated E+E Elektronik EE-03 sensor for environmental humidity and temperature. This sensor has a stated accuracy of +/- 3% relative humidity for environmental conditions between 10-100% RH and a temperature accuracy of +/- 0.3 C at an ambient temperature of 20 C. The S1 collected 10.13 hours of data during LAPSE-RATE.

Beyond the DataHawk2 and S1, CU operated two other aircraft on a more limited basis. The first of these was the X-UAV Talon RPAS (Figure 1c). These systems are small (1.7 m, 3 kg) airframes made of EPO foam and were outfitted with the PixHawk2 autopilot system running the open source Ardupilot Plane software. For LAPSE-RATE, the Talons were set up to fly for approximately 45 minutes on a single battery at a cruise speed of 18 m s$^{-1}$. As with the DataHawk2, this relatively slow flight speed provides high resolution measurements from the custom payload that was flown on this platform. The Talons operated for LAPSE-RATE carried a Vaisala RSS904 sensor (similar sensor module to that used in the Vaisala RS-92 radiosonde) to make measurements of temperature, pressure and humidity. The RSS904 features a capacitive wire temperature sensor with a 0.1 C resolution, a thin-film capacitor humidity sensor with a resolution of 1 %, and a silicon pressure sensor with a measurement resolution of 0.1 hPa, and expected



accuracies of 0.2 C, 2% and 0.4 hPa, respectively, in temperature, RH and pressure. The Talon collected 4.2 hours of data during LAPSE-RATE.

The RSS904 sensor suite was integrated into a custom 3D-printed nosecone. In addition to measurements from the RSS904, wind speed and direction were estimated by the autopilot system, using a combination of the GPS velocities, aircraft attitude measurements from the inertial navigation system and aircraft airspeed. For a limited number of Talon flights, the aircraft carried a "microsonde" custom temperature, pressure and humidity sensor suite developed at CU. The microsonde was developed to be integrated into a Lagrangian drifter called the Driftersonde (Swenson et al., 2019) that could be launched from small RPAS for atmospheric research. The microsonde includes a MS8607 PTH sensor from TE Connectivity, which consists of a piezoresistive pressure sensing element, which measures both barometric pressure and temperature. The piezoresistive Micro-Electro-Mechanical Systems (MEMS) measure atmospheric pressure relative to a vacuum inside the MEMS, sealed by a thin membrane. A Wheatstone bridge logs temperature by measuring the temperature-dependent resistance. Additionally, the MS8607 includes a capacitive sensing element that features a dielectric polymer film, sensitive to humidity, between two electrodes to measure relative humidity. In addition to the MS8607, the microsonde includes a CAM-M8Q GPS module (ublox) that provides reception of up to 3 global navigation satellite systems (GNSS) (GPS, Galileo, GLONASS, BeiDou) at a given time. The integration of the microsonde on the Talon during LAPSE-RATE was, in part, to provide a dataset to allow for direct comparison between the measurements it produced to those from the RSS904, which is considered to be an industry standard sensor suite.

Finally, the other aircraft operated by CU that saw limited action during LAPSE-RATE was the TTwistor RPAS (Figure 1d). The TTwistor is based on an earlier, proven design, the Tempest (cite papers), which was deployed extensively to study severe weather in the midwestern United States. The TTwistor is a dual-motor RPAS capable of completing up to three hours of continuous flight operations with an 18,000 mAh high-voltage Lithium Polymer (Li-HV) battery pack. This results in a range of 120-240 km, depending on operational mode and flight pattern, at a typical cruise speed of 28 m s$^{-1}$. Additionally, the aircraft was designed to be capable of a fast (40+ m s$^{-1}$) dash speed to allow it to get out of situations with strong winds around convective storms. The aircraft carries a PixHawk autopilot system for autonomous navigation between GPS coordinates and remote operation by a ground pilot. For LAPSE-RATE the TTwistor carried a payload consisting of a multi-hole pressure probe (MHP) from Aeroprobe Corporation, a VectorNav VN-200 inertial navigation system (INS), and a Vaisala RSS904 pressure, temperature and humidity sensor system. The VN-200 has a stated dynamic accuracy of 0.1 degrees (RMS) in pitch and roll measurements and 0.3 degrees (RMS) in magnetic heading. Additionally, it features a stated 2.5 m (5.0 m) accuracy in horizontal (vertical) position and 0.05 m s$^{-1}$ accuracy in velocity. In combination the airspeed, angle of attack and sideslip angles from the MHP and attitude and ground-relative velocity information from the VN-200 allow for accurate calculation of the vertical and horizontal winds.

The extended endurance of the TTwistor, along with advanced implementation of autopilot capabilities and a surface tracking vehicle (Figure 1d) allows this platform to be deployed in a "follow-me" mode. In this mode of operation,



the flight team is inside of a moving vehicle and the aircraft is programmed to stay within a given distance of the vehicle while in flight. This allows for the execution of extended horizontal transects for collection of data on spatial variability and gradients. Follow-me has been used to make measurements of outflow boundaries associated with severe storms, and during LAPSE-RATE was implemented to attempt to better understand the influence of surface heterogeneity on the overlying atmospheric state. The TTwistor collected 7.49 hours of data during LAPSE-RATE.

### 3 Description of measurement locations, flight strategies and completed flights/sampling

Over the course of the one-week LAPSE-RATE campaign, the platforms described above conducted flights over a variety of different areas to collect atmospheric data (Figure 2). All of the aircraft involved conducted flights alongside the CU Mobile RPAS Research Collaboratory (de Boer et al., 2020f) at Leach Airport (37.79 N, 106.05 W) to allow for intercomparison of platforms and sensors. Unfortunately, the data from the Talon and TTwistor were not processed in time to be included in the formal intercomparison conducted through this effort (Barbieri et al., 2019).

Besides the limited flights conducted at Leach Airport, the remainder of the CU DataHawk2 LAPSE-RATE operations were conducted at a farm site in the western part of the San Luis Valley (approximately 37.86 N, 106.18 W, 2326 m above mean sea level (MSL)) near the intersection of County Road J and State Route 45 (see Figure 3). This site featured large crop circles, each having different states of irrigation and plant activity. At this site, two primary flight modes were employed. The first of these was a basic profiling mode, where the aircraft conducted repeated vertical ascents and descents while following an orbital flight track. These flights were designed to evaluate the development of the boundary layer and temporal evolution of its structure during the morning hours. Profiles typically ranged between the surface and 500 m above ground level (AGL). In addition to these profiling flights, additional emphasis was placed on attempting to measure differences in boundary layer development over different surface types, mainly between crop circles with different amounts of plant coverage and irrigation levels. For those flights, the aircraft were operated in a loiter pattern over two different crop circles for extended time periods.

With the exception of 19 July, the BST S1 was operated at the mouth of the Penitente Canyon, just to the north of the intersection between routes 38A and 40G (37.84 N, 106.27 W, 2400 m MSL) (Figure 4). At this location, the aircraft conducted nearly continuous profiling flights between the surface and approximately 800 m AGL. Flights were almost exclusively conducted in the morning and early afternoon (local time) hours. Flights on 19 July were conducted just to the north of the town of Villa Grove, Colorado off of private property located off of County Road 57 at the end of an old air strip (38.27 N, 105.94 W, 2415 m MSL). These flights included a series of stepped racetracks between two points situated across a valley from one another and were designed to measure density-driven cold-air drainage flow into the main San Luis Valley. Flight altitudes for these drainage-flow flights spanned between the surface and approximately 350 m AGL.



The Talon conducted scientific flight operations on July 18-19 in two different locations, as shown in Figure 5. Flight operations on 18 July were conducted at Leach Airport, and included three profiling patterns to approximately 700 m AGL. These flights were meant to capture temporal variability of boundary layer structure and extend the vertical extent of the MURC measurements obtained at that location. Additionally, these flights offered an opportunity to compare the Talon observations with those from the MURC instrumentation tower, and some time was spent at the end of each flight operating at lower altitudes in proximity to the MURC. On 19 July, the Talon conducted six research flights in the Saguache Canyon starting around 06:15 local time, with operators perched above the valley floor off of County Road Cc36 (approximate location 38.105 N, 106.242 W). From here, the Talon was flown in a stepped racetrack pattern across the valley to the north-northeast, covering approximately 2 km of horizontal distance and transitioning between different altitudes at the conclusion of each racetrack to attempt to map out the spatial extent of the valley drainage flow. This included sampling at eight different altitudes, ranging between 350 and 30 m AGL, with the first six altitudes separated by 50 m and the last one by 20 m.

Finally, the TTwistor was operated in two different modes during the LAPSE-RATE campaign, with flight locations illustrated in Figure 6. On 18 July, the aircraft conducted flights in "follow-me" mode, where the aircraft operators are positioned in a vehicle-based mobile ground station. Under this mode of operation, the aircraft conducted extensive horizontal transects from Leach Airport through the eastern half of the San Luis Valley. These flights included sampling at various altitudes and were designed to document spatial variability between the more agricultural portions of the valley and the dry eastern shrubland. In total, three "follow-me" flights were conducted, with the earliest of these taking the aircraft directly south of Leach Airport, before completing an easterly transect and returning back to the north. The second "follow-me" flight took the aircraft further to the east, and included a north-south leg approximately ¼ and ¾ of the way through the flight. The third flight followed a similar route to the second. In general, the first half of these "follow-me" flights were conducted at approximately 300 m AGL, with the second half conducted at approximately 120 m AGL. In addition, the first flight also included a short sampling period at approximately 25 m AGL to foster comparison with the MURC tower. On 19 July, the TTwistor operated from the western end of the Saguache Airport to conduct coordinated sampling of cold-air drainage in the Saguache Canyon. From there, the aircraft conducted three flights in which it stepped through altitudes between 45 and 270 m AGL in 45 m increments, while operating in a large orbit. The large orbit was meant to capture spatial variability of the drainage flow within the width of the canyon, with the aircraft covering approximately 1/3 of that width. Flights on the 19th were initiated at around 06:00 local time (MDT) in order to capture the most intense portions of the drainage flow, which begins to taper off with solar heating of the surface.

Some of the primary measurements from these platforms and the flights conducted during LAPSE-RATE are included in Figure 7. These figures show similar observations from the different platforms, with some notable exceptions. First, the S1 shows an influence of heating of the temperature sensors while the aircraft is on the ground. Because of this heating, the S1 temperature sensor provides excessive temperature readings for the first portion of the flight, resulting in super-adiabatic lapse rates extending from the surface to around 100 m altitude. This is resolved after the



aircraft has been in the air for several seconds. The DataHawk2 slow temperature sensor shows similar structures, but their prevalence is greatly reduced if analyzing the coldwire (fast) temperature measurement. Notably, both of these platforms have some variability near the surface that is likely to be a result of measurement error there, rather than atmospheric features near the surface. In addition to these near-surface features, it is also notable that the TTwistor

has substantially more variability in its measured wind speed than the other platforms. It is possible that some of this is the result of measurement error, though it is important to note that the largest excursions are generally associated with flight towards the end of the flight, was completed under manual pilot control and is therefore less stable than the autopilot-controlled flight period.

Flight operations conducted by CU and BST were completed under Certificates of Authorization issued by the US Federal Aviation Administration (FAA). These COAs were developed to support the safe operation of these platforms to 914 m AGL, including alongside a moving control station.

**4 Data Processing and Quality Control**

As discussed in de Boer et al. (2020b), LAPSE-RATE data files are provided in NetCDF format with a common file name structure. This common structure (xxx.ppppp.lv.yyyymmdd.hhmmss.cdf) provides information on the institute generating the data (xxx), the platform used to collect the data (ppppp), the level of data processing (lv), and the date and time that the measurements were obtained. The institutions covered in the current paper include the University of Colorado Boulder (UCB) and Black Swift Technologies (BST), with platform identifiers including the DataHawk

2 (DATHK), the S1 (BSTS1), the Talon (TALN3) and the TTwistor (TWSTR).

The teams provided different levels of processing and quality control for the measurements from these platforms. Processing of the DataHawk2, Talon and TTwistor resulted in files at the b1 level. The S1 files were only processed to the a1 level. Similar limits were applied to all three UCB datasets to provide a b1 level dataset. These values were

selected as physical limits that should not be exceeded for data collected during LAPSE-RATE, and are reviewed in Table 2. As a reminder, the levels that were defined for LAPSE-RATE in general included:

**a0:** Raw data converted to netCDF

**a1:** Calibration factors applied and converted to geophysical units

**b1:** QC checks applied to measurements to ensure that they are "in bounds". Missing data points or those with

bad values should be set to -9999.9

**c1:** Derived or calculated value-added data product (VAP) using one or more measured or modeled data (a0 to c1) as input

Processing of the DataHawk data included calibration of the coldwire temperature and calculation of the winds. The

coldwire voltages were calibrated against temperatures derived from the SHT sensor. This relationship is expected to be linear and the calibration relationship is derived using a first-order polynomial fit. This fit is calculated on a flight-by-flight basis, leveraging temperature measurements from parts of the dataset that were collected while the aircraft



was in flight. In addition, the first 1000 datapoints (100 s at 10 Hz) are also omitted from this calculation to avoid contamination from the surface heating of the temperature sensor discussed earlier. DataHawk2 winds are calculated

by leveraging observations from the pitot tube, the GPS, and the onboard inertial measurement unit (IMU). Because the DataHawk2 winds calculated from these sensors include a periodic signal that corresponds to the frequency of an orbital circle, additional filtering is applied to these winds. This technique applies a 15th order low pass infinite impulse response (IIR) filter to the originally calculated winds to produce a smoothed version. Additionally, using yaw data from the autopilot, the mean wind is calculated over any single orbit. Finally, the high-frequency component of the

originally calculated winds (difference between the calculated wind and the output from the lowpass filter) is added back to the wind calculated across the orbital means. As a result, these wind estimates are meant to primarily provide perspective on the mean wind structure with altitude, in addition to the higher frequency turbulent structure.

The Talon and TTwistor data also require the calculation of wind components. For both datasets, winds are calculated

using formulae available in van den Kroonenberg et al. (2008). This technique takes input from the onboard GPS, IMU and airspeed sensor to calculate three-dimensional wind components. The TTwistor carried a multihole pressure probe to measure angle of attack and sideslip angles, while the Talon only carried a standard Pitot tube for estimating airspeed. As a result the Talon angle of attack and sideslip were estimated to be a constant 1.75 degrees and 0 degrees, respectively, for the entire flight. This assumption results in the omission of fine turbulent motion that would not be

captured without understanding of the angle of airflow over the airspeed sensor. For both platforms, a series of corrections are applied to account for offsets in the calculated true airspeed (TAS), and potential angular offsets between the IMU and the airspeed sensor (yaw and pitch offsets). These offsets are calculated in an iterative manner by minimization of the variance of the calculated winds with an order of TAS, pitch, yaw, TAS and yaw. The last two TAS and yaw corrections are applied to apply a finer scale correction than possible in the first round.


## 5 Data Availability

The data files from the LAPSE-RATE project are generally being archived under a LAPSE-RATE community established at the Zenodo data archive (https://zenodo.org/communities/lapse-rate/). From here, LAPSE-RATE

observations are available for public download and use. Data Object Identifiers (DOIs) were automatically generated by the Zenodo archive at the data version and product level. Data from the different sources described above are posted as individual datastreams on the archive, with each of the platforms described in the previous section having their own DOI. It is important to note that each platform may have several different levels of data available. Therefore, data products with different levels of processing and quality control may be provided with separate DOIs. This means

the files and data described in this publication are spread across a variety of DOIs, and that additional DOIs could be created in the future that include LAPSE-RATE data, as additional data products are developed.

As of the writing of this manuscript, the CU DataHawk2 dataset (de Boer et al., 2020c) is available at Zenodo.org (https://zenodo.org/record/3891620#.X48psy9h1dA) under DOI http://doi.org/10.5281/zenodo.3891620. This



version 2.0 of this dataset includes both a1 and b1 level files, with the b1 files featuring additional quality control and processing of wind data, as described in section 4 above. Data from the CU Talon aircraft includes three versions (de Boer et al., 2020d). The original version had some issues with file names and the timestamp that was included in the files, which have been fixed in subsequent versions. Therefore, users should use the latest available version (currently 3.0), which is available at Zenodo.org (https://zenodo.org/record/4096451#.X48xYy9h1dA) under

http://doi.org/10.5281/zenodo.4096451. The CU TTwistor data (de Boer et al., 2020e) are also available at Zenodo.org (https://zenodo.org/record/4110626#.X48xsC9h1dA) under DOI http://doi.org/10.5281/zenodo.4110626. As with the Talon data, there are multiple versions, and the most current version as of the writing of this manuscript is 3.0. Finally, the BST S1 datasets (Elston and Stachura, 2020) are also available at Zenodo.org (https://zenodo.org/record/3861831#.X48x4i9h1dA) under DOI http://doi.org/10.5281/zenodo.3861831, with the

most current version being 1.1.

## 6 Summary

This manuscript provides an overview of data collected by four different RPAS operated during the 2018 LAPSE-RATE campaign in the San Luis Valley of Colorado. These included the University of Colorado DataHawk2, Talon

and TTwistor aircraft, and the Black Swift Technologies S1. In combination, these vehicles collected over 30 flight hours of meteorological data between 14-20 July, 2018, covering altitudes between the surface and 915 m AGL. Data from these vehicles are available for public download from zenodo.org, and the previous sections document processing conducted on this dataset before publication, as well as information on the expected accuracy of the sensors deployed on these systems. The contributions made by these RPAS and their operating teams to the broader LAPSE-RATE

effort represent a significant component of the measurements obtained to gain understanding on boundary layer processes and phenomena in this high-altitude environment. In combination with surface-based and numerical data, these data are currently being used to evaluate a variety of scientific hypotheses related to the primary measurement objectives of LAPSE-RATE.


**Author Contributions**

GB was the PI for LAPSE-RATE and lead organization of the field campaign. Additionally, he processed data for the DataHawk2, Talon and TTwistor aircraft, and was the primary author of the current manuscript. C.D., S.B., S.S., R.L., K.G., J.E. and M.S. assisted with the processing of the datafiles described in this manuscript. C.D., S.B., D.L,

J.E., D.H., M.S., R.L., S.S., C.C., A.D., A.S., K.G., Z.L. and F.Q. were involved with field operation of the aircraft and data processing during LAPSE-RATE, with D.H. serving as chief pilot for the University of Colorado operations. B.A. and E.F. were the PIs for the Talon and TTwistor aircraft deployed during LAPSE-RATE, while D.L. was the PI for the DataHawk2. All authors contributed to the review and development of the manuscript.

**Acknowledgements**



General support for salary and overhead associated with the collection of these datasets was provided by the NOAA Physical Sciences Laboratory, the University of Colorado's Integrated Remote and In-Situ Sensing (IRISS) grand challenge project, and Black Swift Technologies. We would additionally like to recognize financial support for student participation and travel from the National Science Foundation (NSF AGS 1807199) and the US Department of Energy (DE-SC0018985). General support for the LAPSE-RATE campaign was provided by the International Society for Atmospheric Research using Remotely-piloted Aircraft (ISARRA).

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





| Platform | Measurement | Sensor | Stated Accuracy |
|---|---|---|---|
| *CU DataHawk2* | Temperature (C) | Sensiron SHT-31 | +/- 0.2 C (0-90 C) |
| | Relative Humidity (%) | Sensiron SHT-31 | +/- 2% (0-100%) |
| | Pressure (hPa) | | |
| | Wind Speed (m s$^{-1}$) | Multiple | Undetermined |
| | Wind Direction (m s$^{-1}$) | Multiple | Undetermined |
| | Up- and Downward Looking Brightness Temperature (C) | Semitec 10TP583T | Undetermined |
| | | | |
| *CU Talon* | Temperature (C) | Vaisala RSS904 | 0.2 C |
| | Relative Humidity (%) | Vaisala RSS904 | 2% |
| | Pressure (hPa) | Vaisala RSS904 | 0.4 hPa |
| | Wind Speed (m s$^{-1}$) | Multiple (from PixHawk autopilot) | Undetermined |
| | Wind Direction (m s$^{-1}$) | Multiple (from PixHawk autopilot) | Undetermined |
| | | | |
| *CU TTwistor* | Temperature (C) | Vaisala RSS904 | 0.2 C |
| | Relative Humidity (%) | Vaisala RSS904 | 2% |
| | Pressure (hPa) | Vaisala RSS904 | 0.4 hPa |
| | Wind Speed (m s$^{-1}$) | Multiple, including Aerosonde MHP | Undetermined |
| | Wind Direction (m s$^{-1}$) | Multiple, including Aerosonde MHP | Undetermined |
| | | | |
| *BST S1* | Temperature (C) | E+E Elektronik EE-03 | +/- 0.3 C (at 20 C) |
| | Relative Humidity (%) | E+E Elektronik EE-03 | +/- 3% (at 21 C) |
| | Pressure (hPa) | BST MHP | |
| | Wind Speed (m s$^{-1}$) | Multiple, including BST MHP | Undetermined |
| | Wind Direction (m s$^{-1}$) | Multiple, including BST MHP | Undetermined |
| | | | |

**Table 1:** An overview of the platforms and meteorologically-relevant measurements from each.



| Variable | DataHawk2 | S1 | Talon | TTwistor |
|---|---|---|---|---|
| Temperature (C) | -50<T<50 | N/A | -50<T<50 | -50<T<50 |
| RH (%) | 0<RH<110 | N/A | 0<RH<110 | 0<RH<110 |
| Pressure (hPa) | 0<p<1000 | N/A | 0<p<1000 | 0<p<1000 |
| Altitude (m MSL or AGL) | 0<alt<5000 | N/A | 0<alt<5000 | 0<alt<5000 |
| Latitude (deg) | 30<lat<40 | N/A | 30<lat<40 | 30<lat<40 |
| Longitude (deg) | -120<lon<-100 | N/A | -120<lon<-100 | -120<lon<-100 |
| u, v (m s$^{-1}$) | -50<u,v<50 | N/A | -50<u,v<50 | -50<u,v<50 |
| w (m s$^{-1}$) | -20<w<20 | N/A | -20<w<20 | -20<w<20 |
| Wind Speed (m s$^{-1}$) | 0<wspd<100 | N/A | 0<wspd<100 | 0<wspd<100 |

**Table 2:** Quality control limits applied to the different data b1 datasets. To date, S1 data are only available as a1 level files.

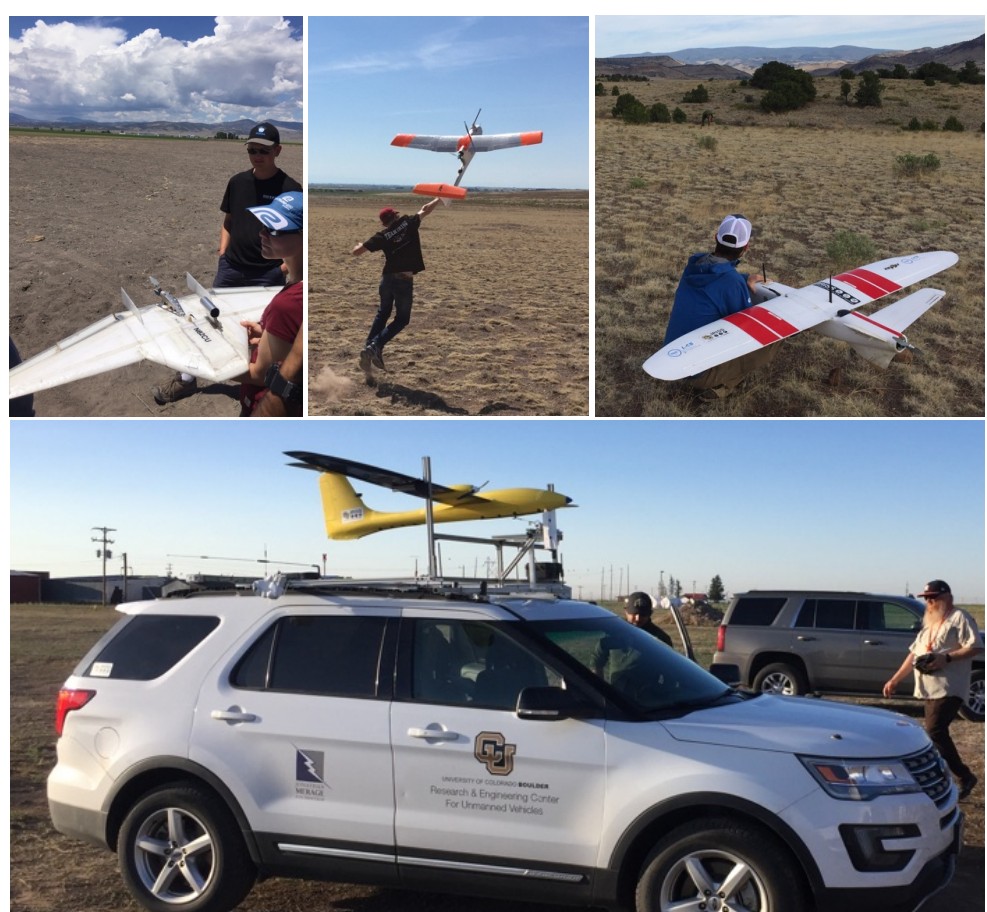

**Figure 1:** The aircraft used to collect the datasets described in this manuscript. Clockwise from top left: The CU DataHawk2, the BST S1, the CU Talon, and the CU TTwistor mounted on top of a tracker vehicle for launch prior to "follow me" operations.


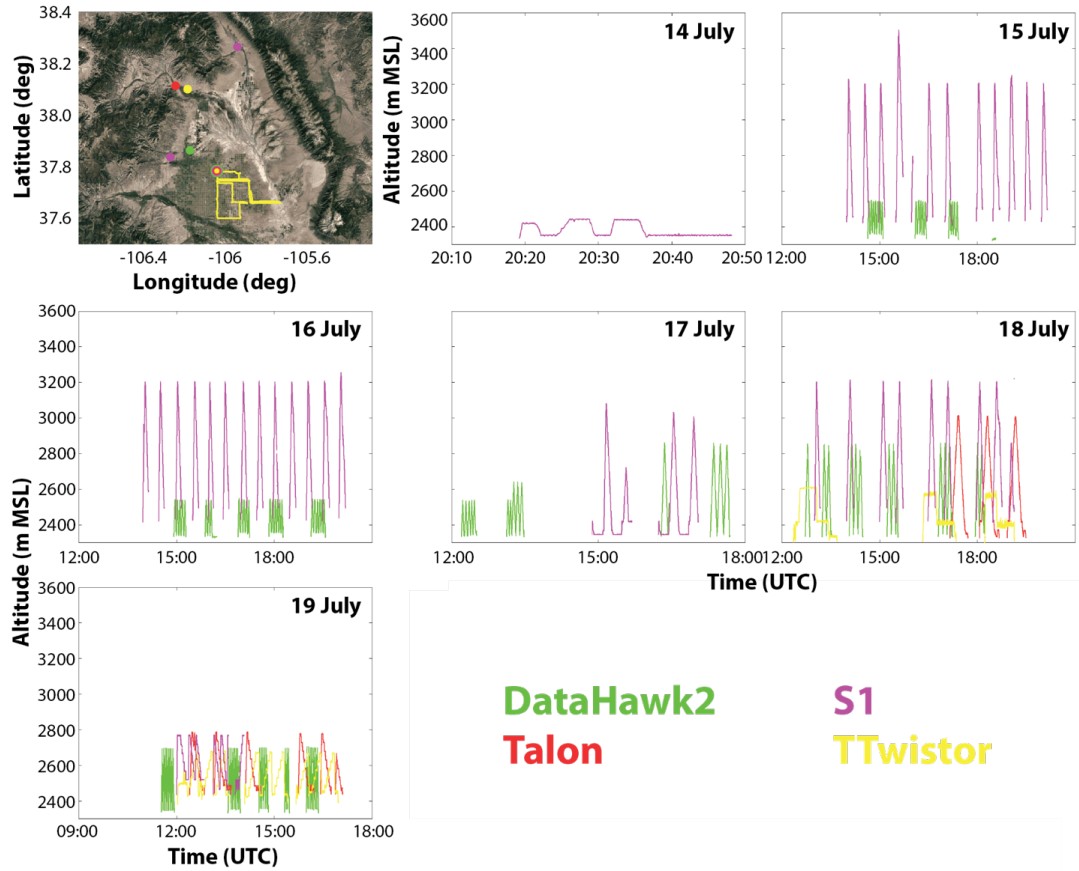

**Figure 2:** The flight locations for each aircraft (left) and altitudes covered by the different aircraft broken down by day (right). Map background is from © Google Maps, downloaded through their API.


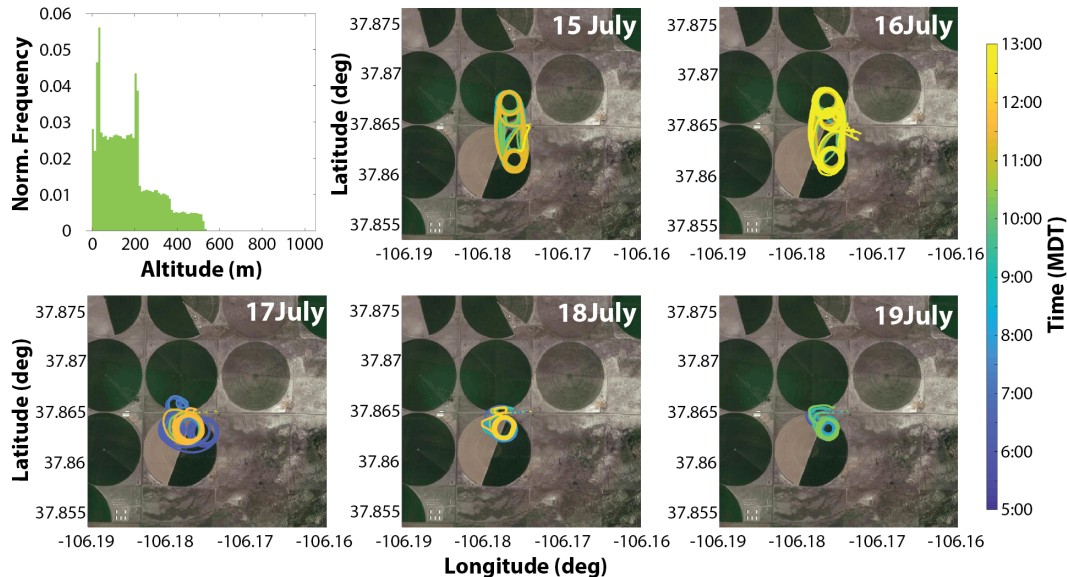

**Figure 3:** A histogram of altitudes covered during LAPSE-RATE by the DataHawk2 (top left), along with the flight patterns covered by this aircraft during individual days of the campaign. See the map in Figure 2 for additional details on flight location. Map backgrounds are from © Google Maps, downloaded through their API.


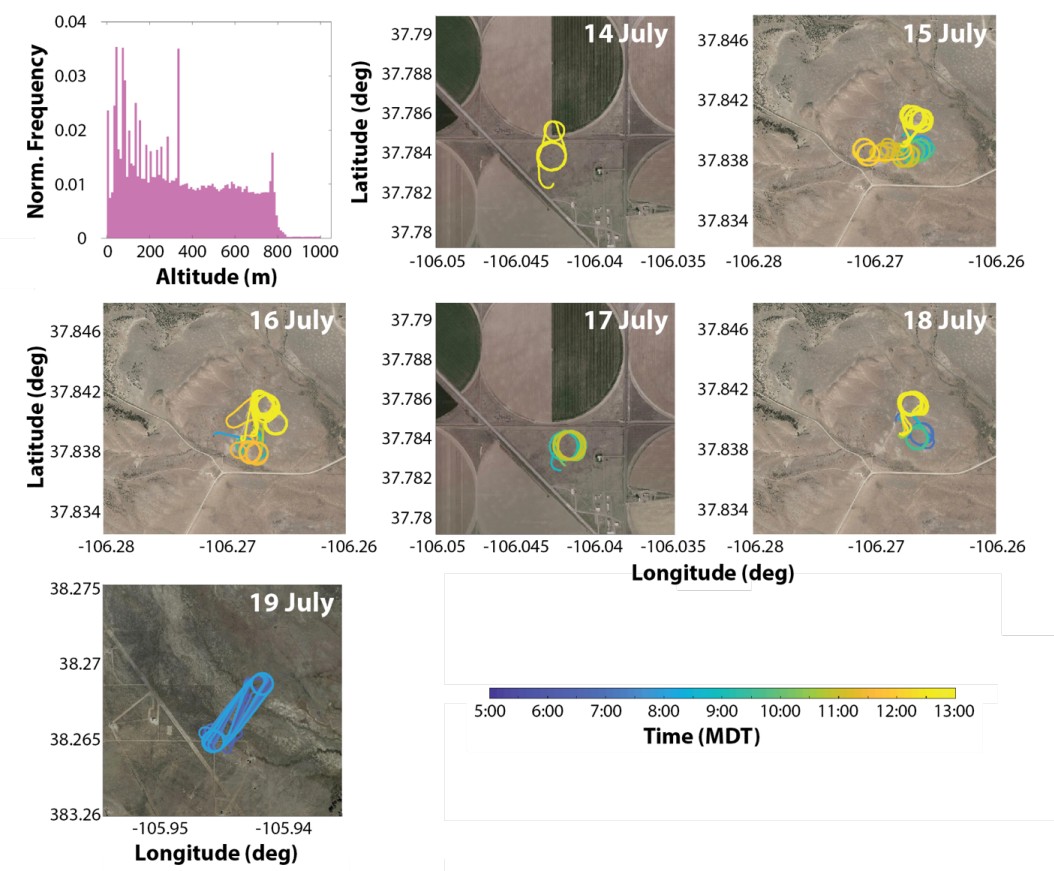

**Figure 4:** A histogram of altitudes covered during LAPSE-RATE by the S1 (left), along with the flight patterns covered by this aircraft during individual days of the campaign. See the map in Figure 2 for additional details on flight locations. Map backgrounds are from © Google Maps, downloaded through their API.

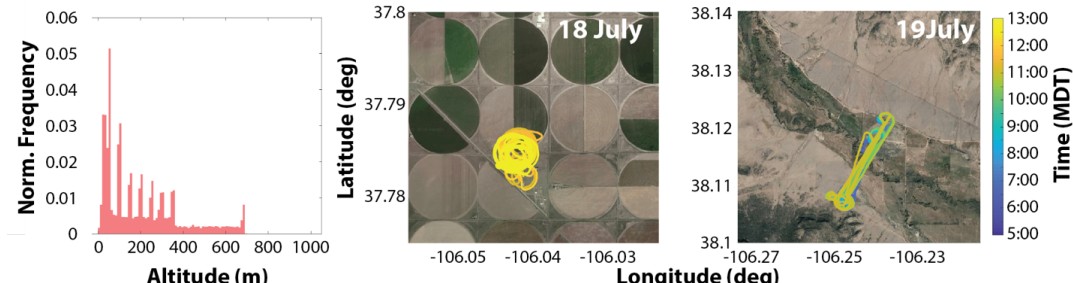


**Figure 5:** A histogram of altitudes covered during LAPSE-RATE by the Talon (left), along with the flight patterns covered by this aircraft during individual days of the campaign. See the map in Figure 2 for additional details on flight locations. Map backgrounds are from © Google Maps, downloaded through their API.


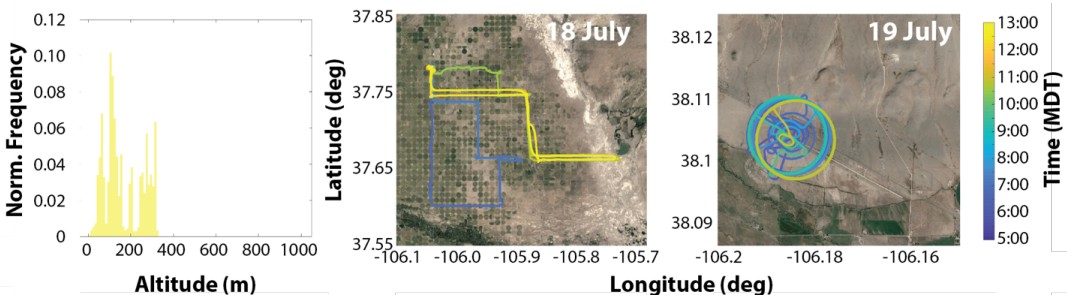

**Figure 6:** A histogram of altitudes covered during LAPSE-RATE by the TTwistor (left), along with the flight patterns covered by this aircraft during individual days of the campaign. See the map in Figure 2 for additional details on flight locations. Map backgrounds are from © Google Maps, downloaded through their API.


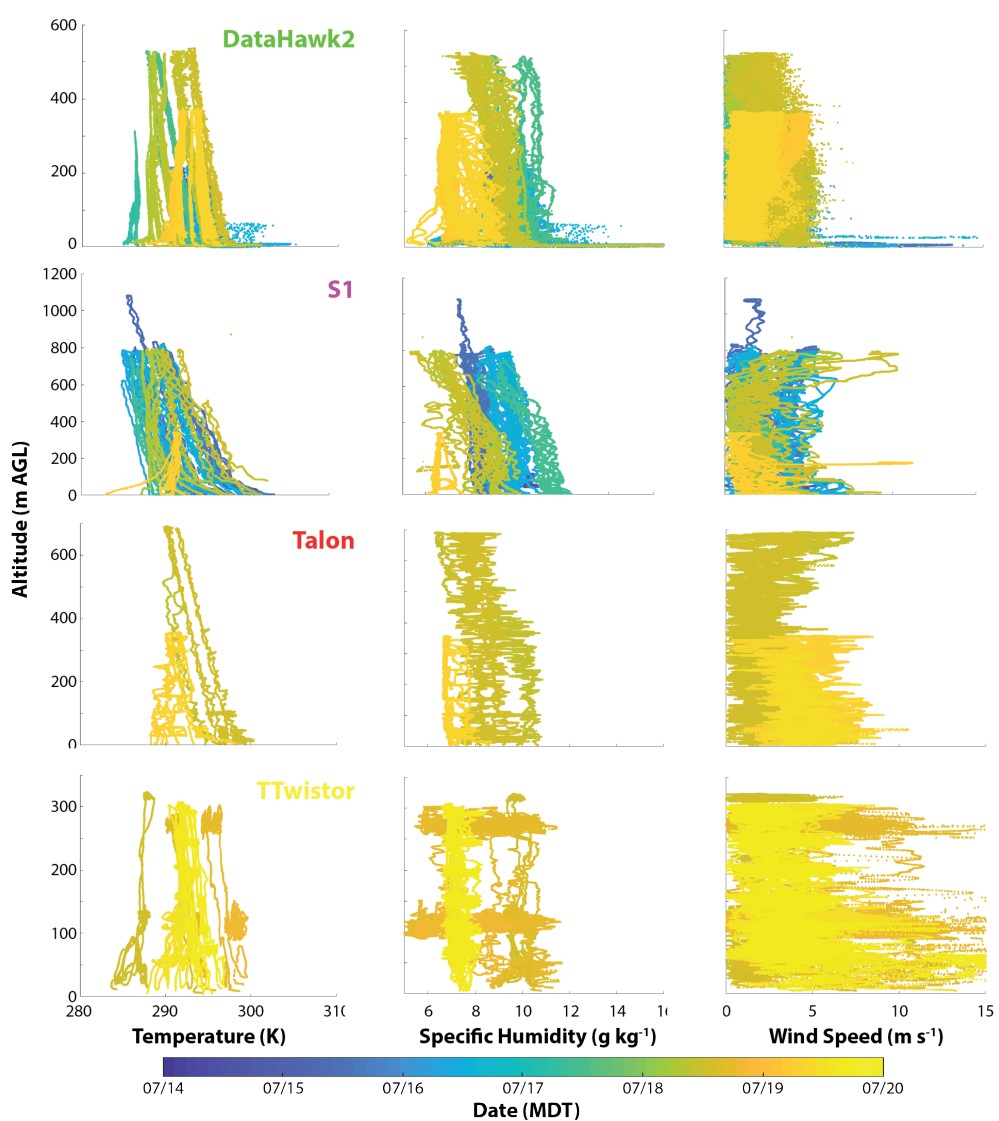

**Figure 7:** Profiles of (left to right) temperature, specific humidity and wind speed from the DataHawk2, S1, Talon and TTwistor (top to bottom). Colors indicate the date of flight.
