# Peer review of "University of Colorado and Black Swift Technologies RPASbased measurements of the lower atmosphere during LAPSE-RATE"

_Earth System Science Data, 2020_

## Referee Comment (RC1) · Anonymous Referee #1 · 17 Dec 2020

The authors provide an overview of the data acquired using four different remotely piloted systems operated by University of Colorado and Black Swift Technologies over the one week period of LAPSE-RATE. The article is largely comprised of a general overview of the aircraft and flights that contribute to the dataset referenced in the manuscript. As such, the manuscript provides sufficient detail to understand the contents of the dataset. Although I would have preferred more information about the uncertainty of the measured quantities, the data quality control checks are provided, which provides a broad sense of the data confidence level. I therefore recommend publication in ESSD. However, I do have a few minor comments that the authors may wish to address before publication: 1) Little-to-no information is provided about the data acqui-

sition systems used by the aircraft. At the very least, some indication as to the data acquisition rates should be included, particularly given the scatter shown in Figure 7. Corresponding to this point, I would also have liked to have seen some reference to the different response times/frequency response of the sensors. A broad range of sensors are used, and having some knowledge of the time response of the systems relative to the data acquisition rate is important when utilizing the provided data. 2) There is a typo on page 5: "(cite papers)" should be replaced with the relevant citations. 3) The winds up to 15 m/s measured by the TTwistor shown in Figure 7 seem quite high. Are these values comparable to the winds measured by other LAPSE-RATE systems? If so, some commentary should be provided as to why they are so much higher than the other aircraft (e.g. Talon) which were measuring at the same time.

---

## Referee Comment (RC2) · Anonymous Referee #2 · 10 Jan 2021

General comments:

The manuscript provides a concise but comprehensive summary of the measurement priorities and methods adopted by one of the teams participating in the unique LAPSE-RATE measurement campaign. The paper provides informative descriptions of the mission (sensing) and operational (performance) capabilities of four remotely piloted aircraft that were used to sample the lower atmosphere over an extended period, targeting a number of science goals enumerated in the text. The paper closes with an overview of sampling times and locations and a summary plot of the data that were collected.

Specific comments:

Following are a few comments that may help improve the paper's impact.

While measurement data are shown in Figure 7, their presentation and analysis is not a primary focus of the paper, so the title is a little bit misleading. Perhaps the title should emphasize "measurement priorities and methods" rather than simply "measurements."

Table 1 provides a helpful summary of mission (sensing) capabilities. It might be nice to also include a table of operational (performance) capabilities and outcomes (e.g., campaign flight hours, air-relative distance traveled, etc) as well as tasking (e.g., in the context of the measurement priorities listed in Lines 71-75). Much of this information is already provided in the narrative, but a table would facilitate comparisons between platforms.

In providing attribution to Google maps in the captions for Figures 3-6, you might briefly note whether the imagery is consistent with the state of the terrain at the time of the experiments. (E.g., were crops growing as indicated? What was their state of maturity?) This information may be important to someone using the data.

Figure 7 is a nice summary of key data that were collected. The given format is very sensible, with each column depicting a particular measurement variable on a common scale, each row representing the measurements obtained from a given aircraft, and the color map representing the time during the campaign when the data were collected. I might also suggest adopting the same altitue scale for all of the plots, regardless of the platform. This will compress the data for all but the S1, and will generate a lot of white space, but it will also make the rows of data more directly comparable. For example, one would expect to see a fairly consistent lapse rate slope in the temperature data.

Technical corrections:

Line 171: An editorial note ("cite papers") still appears in the manuscript.

---

## Author Comment (AC1) · 4 Mar 2021

**Author Responses to Referee and Topical Editor Comments for essd-2020-333** "University of Colorado and Black Swift Technologies RPAS-based measurements of the lower atmosphere during LAPSE-RATE"

Referee comments are provided in black font, and author responses are included in blue font. Additionally, we have included a "tracked changes" version of the manuscript at the end of our responses to provide information on what has been changed.

**Anonymous Referee #1**

The authors provide an overview of the data acquired using four different remotely piloted systems operated by University of Colorado and Black Swift Technologies over the one week period of LAPSE-RATE. The article is largely comprised of a general overview of the aircraft and flights that contribute to the dataset referenced in the manuscript. As such, the manuscript provides sufficient detail to understand the contents of the dataset. Although I would have preferred more information about the uncertainty of the measured quantities, the data quality control checks are provided, which provides a broad sense of the data confidence level. I therefore recommend publication in ESSD. However, I do have a few minor comments that the authors may wish to address before publication:

We appreciate the time provided by the reviewer to provide this feedback and improve the quality of the manuscript.

1) Little-to-no information is provided about the data acquisition systems used by the aircraft. At the very least, some indication as to the data acquisition rates should be included, particularly given the scatter shown in Figure 7.

We have included data acquisition rate information for all four aircraft in the updated manuscript. Please note that some of the scatter shown in Figure 7 (particularly related to TTwistor winds) was the result of a transition to manual flight. Those data have been removed in the updated version of the figure since these short portions of the flight were substantially more erratic than the autopilot flight, challenging the wind estimation algorithms.

2) Corresponding to this point, I would also have liked to have seen some reference to the different response times/frequency response of the sensors. A broad range of sensors are used, and having some knowledge of the time response of the systems relative to the data acquisition rate is important when utilizing the provided data.

We appreciate this request and completely agree that in theory this information would be helpful at times in analyzing the collected data. Having said this, while we could provide some response times as provided by the manufacturers, much of the real response of the measured quantity is a result of the integration onto the airframe. To this end, these characteristics had not been evaluated fully prior to the LAPSE-RATE campaign, and we lack the ability to do so now given the evolution of sensors and aircraft over the last two and a half years. For our current aircraft

platforms we are doing a better job of fully characterizing the performance of the sensors as integrated on different platforms, but unfortunately such details are not available for the equipment as deployed for LAPSE-RATE.

3) There is a typo on page 5: "(cite papers)" should be replaced with the relevant citations.

That's embarrassing!  The citations have been inserted.

4) The winds up to 15 m/s measured by the TTwistor shown in Figure 7 seem quite high. Are these values comparable to the winds measured by other LAPSE-RATE systems? If so, some commentary should be provided as to why they are so much higher than the other aircraft (e.g. Talon) which were measuring at the same time.

Thank you for pointing this out.  As had been pointed out in the text, we found that the vast majority of the higher wind speeds measured by the TTwistor occurred during times when the pilot was manually flying the aircraft, almost exclusively right after take-off and right before landing.  For clarity, we have removed those points from figure 7.  You may note that there are still a few instances of higher wind speeds measured from TTwistor on 7/18.  These observations were collected during afternoon "follow-me" legs, where the aircraft was flying through some gusty winds.  We have reworded the text that discussed the TTwistor winds to align with how they are now presented.

**Anonymous Referee #2**

General comments:

The manuscript provides a concise but comprehensive summary of the measurement priorities and methods adopted by one of the teams participating in the unique LAPSE- RATE measurement campaign. The paper provides informative descriptions of the mission (sensing) and operational (performance) capabilities of four remotely piloted aircraft that were used to sample the lower atmosphere over an extended period, targeting a number of science goals enumerated in the text. The paper closes with an overview of sampling times and locations and a summary plot of the data that were collected.

We would like to thank the reviewer for the time that they dedicated to reading through the manuscript and offering suggestions for improvement.

Specific comments:

Following are a few comments that may help improve the paper's impact.

While measurement data are shown in Figure 7, their presentation and analysis is not a primary focus of the paper, so the title is a little bit misleading. Perhaps the title should emphasize "measurement priorities and methods" rather than simply "measurements."

We appreciate this recommendation.  However, given that the scope of a data paper as presented in ESSD should specifically not include scientific analysis, we feel that use of the word "measurements" in the current title offers a clear perspective on what is provided in this article (rather than, say, model data).   Therefore we would respectfully like to keep the original title.

Table 1 provides a helpful summary of mission (sensing) capabilities. It might be nice to also include a table of operational (performance) capabilities and outcomes (e.g., campaign flight hours, air-relative distance traveled, etc) as well as tasking (e.g., in the context of the measurement priorities listed in Lines 71-75). Much of this information is already provided in the narrative, but a table would facilitate comparisons between platforms.

We appreciate the notion of providing points of comparison between the platforms, and have added the campaign flight hours to Table 1, as recommended.  We're not sure what the reader would gain from including the "air-relative distance traveled", as the aircraft covered varying flight patterns.  While it might be interesting to compare this, it is difficult to understand how such a number would be used scientifically, and, perhaps easier to understand how it may confuse the reader into thinking that this is the ground distance covered by the aircraft relative to a fixed location.  Therefore, we have decided not to include this specific quantity.  Additionally, the tasking information is already outlined in the text, and somewhat difficult to succinctly fit into a table.

In providing attribution to Google maps in the captions for Figures 3-6, you might briefly note whether the imagery is consistent with the state of the terrain at the time of the experiments. (E.g., were crops growing as indicated? What was their state of maturity?) This information may be important to someone using the data.

Thank you for this suggestion.  We agree that such information would be useful.  We did not spend a significant amount of time documenting the environment during the campaign.  However, we do provide some details on the meteorological conditions and the general layout and land usage in the San Luis Valley in the overview article that is included in the special issue that will encompass this article.  We have added some text to the introduction to point readers to that article if interested in more detail on the sampling environment.

Figure 7 is a nice summary of key data that were collected. The given format is very sensible, with each column depicting a particular measurement variable on a common scale, each row representing the measurements obtained from a given aircraft, and the color map representing the time during the campaign when the data were collected. I might also suggest adopting the same altitude scale for all of the plots, regardless of the platform. This will compress the data for all but the S1, and will generate a lot of white space, but it will also make the rows of data more

directly comparable. For example, one would expect to see a fairly consistent lapse rate slope in the temperature data.

We appreciate this suggestion. Because the altitudes are presented as above ground level (AGL) and all of the platforms operated at different elevations from one another and sometimes even from day to day, it is not clear that the profiles would be any more comparable. We could potentially standardize the altitudes to a common (MSL) scale, but, as the reviewer points out this would result in a lot of white space and also limit the amount of detail visible in the observed quantities. Therefore, after consideration, we have decided to keep the altitude scales as they are.

Technical corrections:

Line 171: An editorial note ("cite papers") still appears in the manuscript.

That's embarrassing! The citations have been inserted.

**Topical Editor Comments**

The paper summarizes the thermodynamic and kinematic data sets from four fixed-wing platforms operated by the University of Colorado Boulder (UCB) and Black Swift Technologies (BST) in support of the 2018 LAPSE-RATE campaign. Locations, times and flight patterns for each day's observations are presented, along with discussion of data processing and comments from preliminary evaluation.

We very much appreciate the time dedicated by the editor to recommend these edits and support the overall improvement of the manuscript.

pg 1, lines 35-36: references look like they should be 2020c, 2020d, and 2020e (not a, b, and c)

That is correct – thank you for catching this. We have updated the references accordingly.

pg 2, line 62: reference description doublecheck? From the title, the reference is hurricane observation with UAS (Cione), for the text, "to understand turbulence across the world."

This is the correct reference (the Cione manuscript discusses turbulent flux estimation with the Coyote), but we understand the confusion. We updated the text to include a reference to turbulence in tropical cyclones.

pg 2, line 64: there are two Elston 2011 references, but need a and b added (see lines 441 2011a and 447 2011b). Also these Elston refs have an Elston 2015 between them in the listing.

Thank you for catching this. We have added the a and b in the list of references, reordered them to both fall ahead of the 2015 reference, and included the appropriate a and b labels in other locations in the manuscript where 2011 Elston references were included.

pg 3-5, lines 95, 126, 143, 171, and 187 (maybe other places): refer to subplots of Figure 1 as a, b, c, and d. Suggest these letters should be added to the figure.

We have updated the figure to include the letters.

pg 3, line 99: de Boer et al 2018b is not in the ref list (unless it should be 2018).

Thank you for pointing this out – we have removed the "b" from the reference.

pg 4, lines 140-141, 149, others and in Tables 1 and 2: is it standard meteorological notation to us 20 C rather than 20 °C?

We have typically used C instead of °C, but will leave it up to the ESSD editorial and typesetting team to provide information on what is preferable.

pg 5, line 171: "(cite papers)" to be completed

That's embarrassing! The citations have been inserted.

pg 5, line 183-184: Table 1 is not cited in the text; this might be a good place. Also, here "for accurate calculation of the vertical and horizonal winds." leaves the reader wanting to know how it is calculated and how accurate. Table 1 indicates "undetermined" for all wind measurements.

We included a reference to Table 1 earlier in the section, pointing the reader towards it in the first paragraph. Regarding the wind, we have not completed a full accuracy assessment of our wind estimation, which is why the table indicates "undetermined". We have changed the "accurate calculation" statement in lines 183-184 to "fully-informed calculation", as the alpha and beta angles provided by the probe and the attitude information from the VN-200 offer the full suite of measurements required for wind estimation, as compared to techniques applied to the Talon and DataHawk, which do not carry a MHP.

pg 6, line 198: Should define CU Mobile RPAS Research Collaboratory as MURC for references on pg 7.

We have updated this line to include MURC, and have replaced "RPAS" with "UAS" here, to correctly align with the MURC acronym.

pg 7, line 252: Is "covering approximately 1/3 of the width" centered in the canyon or offset to the West?

We updated the text to reflect an offset. The offset is more towards the northern half of the valley (which generally runs east-west), rather than the western offset referred to by the editor.

pg 8, line270: Suggest adding CoA #s.

We appreciate the suggestion, but are not sure what benefit the reader would gain from including this information, so have not done so.

pg 8, line 292: Should "c1" be b1 here, or are some c1 data sets derived from other c1 data sets?

c1 is correct in this instance – as you state, sometimes it may be that a c1 data product is derived using inputs from another c1 data product.

pg 9. line 299: How are the winds calculated? Is this also by the van den Kroonenberg 2008 reference?

We have included additional information on how the DataHawk autopilot initially calculates the winds, prior to the application of any additional filtering.

pg 9, lines 318-319: Is there a reference with details? Or dissertation?

There is not yet a reference available for this technique, unfortunately.

pg 14, Table 1: This Table is not referenced in the text.

As mentioned above, we have included a reference to Table 1 in section 2.

pg 16, Figure 1: Text calls out Figure 1a, etc., so add a, b, c, and d here in caption or sub-images.

As mentioned above, we have added A-D in the image.

pg 21, Figure 6 left plot: The yellow color is not visible here (ok in Figure 7, though, with more density).

Interesting – it does show up well on my end. I would like to keep it if possible (currently consistent with Figure 2, but can update it if the typeset version is not visible for some reason.

pg 22, Figure 7: With 12 subplots, multiple days, number of flights and the range of the altitude scales, the surface temperature effects described in the paper are difficult for the reader to discern. Could these be pulled out with a zoomed altitude scale in a separate figure?

Consistent with the scope of an ESSD article, we are simply attempting to provide an overview of the observations collected. While we could produce another figure with a consistent zoomed in altitude scale in the lower atmosphere, we fear that justification for doing so would then require

additional analysis of the observations in that area, which, in turn would like extend beyond the scope of what should be included with a data paper.  One of the reviewers recommended making the altitude scales the same for these figures, though doing so would make it difficult to see some of the limited details currently visible.  Additionally, since the altitudes are all AGL and the flight start elevations were different from platform-to-platform and day-to-day, it is not clear to us that standardizing them would be beneficial to the reader.  Therefore, we would like to keep this figure as is.

pg 22, Figure 7: Also, the AGL scale in Figure 7 overlays measurements from different locations with different MSL altitudes. Perhaps a comment relating this in the text or caption should be added?

Thank you for this suggestion – we have included a sentence to the figure caption to indicate that the ground level is different for different platforms (and even for different flights for the same platform).

[revised manuscript text omitted]
). Leveraging information from these sensors, the autopilot computes wind in inertial coordinates as the vector difference of the velocity of the vehicle relative to the ground, minus the velocity of the vehicle relative to the air. The later vector is estimated in magnitude as the airspeed from the pitot-static sensor, and in direction as the orientation of the vehicle longitudinal axis in inertial coordinates. This assumes that the vehicle is pointed into the relative wind (i.e. with only small deviations in sideslip and angle of attack), which is thought to be a good assumption for wind variations with timescales longer than about 1 sec for a vehicle of this scale. 
[revised manuscript text omitted]

**Page 15: [1] Deleted**                **Gijs de Boer**                **2/24/21 11:05:00 AM**